

# The impact of choosing words carefully: an online investigation into imaging reporting strategies and best practice care for low back pain

Emma L. Karran[1,2], Yasmin Medalian[1], Susan L. Hillier[1] and G. Lorimer Moseley[1,3]

[1] School of Health Sciences, University of South Australia, Adelaide, SA, Australia
[2] Royal Adelaide Hospital, Adelaide, South Australia, Australia
[3] Neuroscience Research Australia, Sydney, New South Wales, Australia

Corresponding author
G. Lorimer Moseley,
lorimer.moseley@unisa.edu.au,
lorimer.moseley@gmail.com

## ABSTRACT

**Background.** Low back pain clinical practice guidelines consistently recommend against the routine ordering of spinal imaging; however, imaging is frequently requested in primary care, without evidence of benefit. Imaging reports frequently identify degenerative features which are likely to be interpreted as 'abnormal', despite their high prevalence in symptom-free individuals. The aim of this study was to investigate whether post-imaging back-related perceptions are influenced by providing prior information about normal findings, and to compare the effect of receiving imaging results with best practice care (without imaging). The impact of introducing novel, 'enhanced' reporting strategies was also explored.

**Methods.** This study was a simulated-patient, randomised, multiple-arm experiment. Patient scenarios were presented to volunteer healthy adult participants via an online survey. In the scenarios, 'virtual' patients with low back pain were randomised to one of three groups. Group 1 received imaging and was pre-informed about normal findings. Group 2 received imaging (without pre-information). Group 3 received best practice care: quality information without imaging. Group 1 was further divided to receive either a standard report, or an 'enhanced' report (containing altered terminology and epidemiological information). The primary outcome was back-related perceptions (BRP), a composite score derived from three numeric rating scale scores exploring perceptions of spinal condition, recovery concerns and planned activity. The secondary outcomes were satisfaction and kinesiophobia.

**Results.** Full data were available from 660 participants (68% female). Analysis of covariance revealed a significant effect of group after controlling for baseline BRP scores ($F_{(2,74)} = 10.4, p < 0.001, \eta_p^2 = .04$). Pairwise comparisons indicated that receiving best practice care resulted in more positive BRPs than receiving imaging results, and receiving prior information about normal findings had no impact. Enhanced reporting strategies also positively impacted BRPs ($F_{(1,275)} = 13.06, p < 0.001, \eta_p^2 = .05$). Significant relationships between group allocation and both satisfaction ($F_{(2,553)} = 7.5, p = 0.001, \eta_p^2 = .03$) and kinaesiophobia ($F_{(2,553)} = 3.0, p = 0.050, \eta_p^2 = .01$) were found, with statistically significant pairwise comparisions again in favour of best-practice care.

**Conclusion**. Intervention strategies such as enhanced reporting methods and the provision of quality information (without imaging) have the potential to improve the outcome of patients with recent-onset LBP and should be further considered by primary care providers.

## INTRODUCTION

Low back pain (LBP) is a highly prevalent condition which is most often self-limiting and does not require investigations (*Maher, Underwood & Buchbinder, 2016*). Current clinical practice guidelines for the management of recent-onset LBP outline specific circumstances in which spinal imaging should be considered (*Chou et al., 2011*; *Koes et al., 2010*)—but otherwise recommend *against* routine imaging. Despite this guideline consensus, spinal imaging is frequently requested by primary care providers (*Dagenais, Galloway & Roffey, 2014*; *Williams et al., 2010*); most often inappropriately (*Emery et al., 2013*) and without evidence of benefit (*Chou et al., 2009*).

General Practitioners (GP)s order spinal imaging for reasons including insufficient time to discuss the risks and benefits of scans, concerns regarding their vulnerability for malpractice and a desire to meet patient expectations (*Sears et al., 2016*). Imaging is also often requested for the purpose of providing patients with reassurance (*Howard & Wessely, 1996*). In principle, providing reassurance to patients presenting to primary care with recent-onset low back pain is a consistent recommendation in care guidelines (*Koes et al., 2010*). However, while there is evidence for the reassuring value of medical investigations for some conditions (*Devcich et al., 2012*; *Howard et al., 2005*), the reassuring potential of imaging for patients with LBP has not been demonstrated (*Rolfe & Burton, 2013*).

The failure of spinal imaging to reduce patient concern is hardly surprising when the high prevalence of degenerative changes observed on the scans of asymptomatic adults is considered (*Brinjikji et al., 2014*). Descriptions of these 'normal' changes in imaging reports (using terms suggestive of structural deterioration) are likely to have unwarranted and unnecessary effects on patient perceptions and behaviour (*Sloan & Walsh, 2010*).

In 1998, *Roland & Van Tulder (1998)* suggested that: "Radiologists must take some responsibility for the way their reports are used and interpreted" (p. 230) in order to reduce the potential for harmful *mis*interpretation of imaging findings. A recent study (*McCullough et al., 2012*) has retrospectively examined the effect of including epidemiological information (i.e., a statement reporting the prevalence rates of common imaging findings in asymptomatic adults) alongside lumbar MRI reports, with some promising indications. Also of interest is evidence that providing patients with information about normal cardiac test results—prior to their receipt of their own results—may optimise the reassuring potential of normal test findings (*Petrie et al., 2007*).

In this online study, we investigated varied approaches to imaging reporting and their influence on patient perceptions regarding the condition of their back, concerns about

recovery and plans to engage in activity. We also compared the reassuring value of receiving spinal imaging with the delivery of quality information *without* imaging, consistent with guideline-based 'best practice' care.

The primary aims of this study were:

i. To investigate whether post-imaging back-related perceptions are influenced by the provision of prior information about normal findings;

ii. To compare the effect of receiving imaging results with the effect of receiving best practice care that does not include imaging.

The secondary aims of this study were:

i. To investigate the effect of receiving imaging results compared to best practice care (that does not include imaging) on kinesiophobia;

ii. To investigate whether patients with back pain who receive best practice care from their GP are as satisfied as those who receive spinal imaging;

iii. To compare back-related perceptions following standard reporting of imaging findings with back-related perceptions following 'enhanced' reporting.

## METHODS

This study was approved by the University of South Australia Human Research Ethics Committee (ID 0000035363). A study protocol was registered on Open Science Framework prior to completion of data collection (https://osf.io/4axy6/). Any deviations from that protocol are noted in this manuscript. The reporting of this study is consistent (wherever possible) with the CONSORT 2010 checklist (http://www.consort-statement.org/).

### Design and overview

We used a simulated patient, 3-armed, randomised online experiment to investigate the impact of varied imaging reporting strategies and GP-delivered best practice care on patient cognitions, planned behaviour and satisfaction with healthcare. Adult volunteers were presented with an age-matched hypothetical patient scenario and requested to respond to survey questions *as if they were the patient* described. Participants were randomly allocated to the three study groups (via the survey software randomisation feature) and blinded to conditions other than their own.

### Participants

We recruited participants via an online strategy involving email (using personal and professional networks of the authors), social media and website advertising. Participants were invited to "opt-in" to the study by following a link to the study information and confirmed consent through voluntarily commencing the survey. Individuals were eligible for inclusion if they were aged over 18 years and had sufficient English language proficiency to complete the questionnaire.

### Procedure

Participants used their personal computers or smartphones to access the survey via web-based survey software. They completed a 4–item demographic questionnaire and six
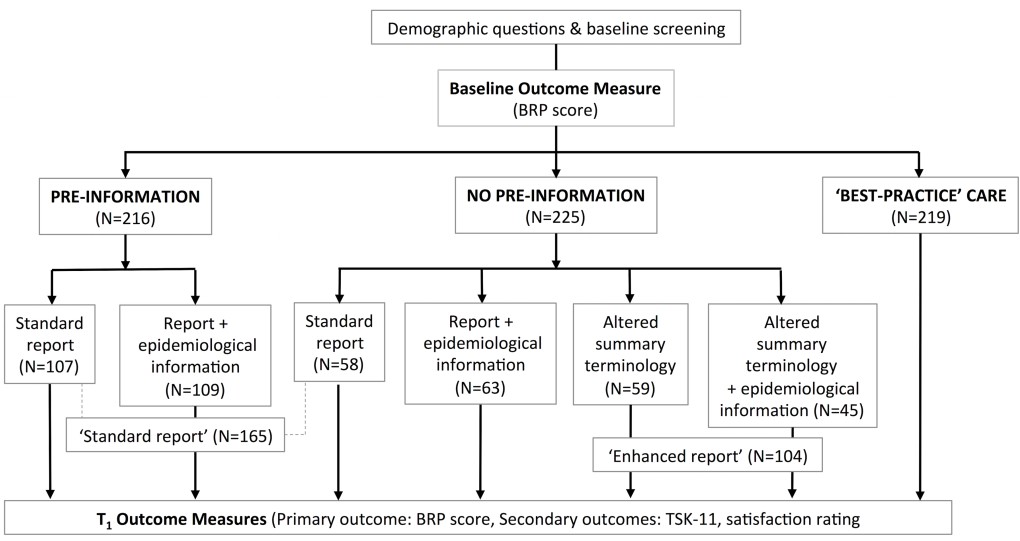

**Figure 1  Online study set-up.**

baseline screening questions, and were provided with a scenario describing a person who injures their back while lifting. Participants were asked to imagine that they were person described in the scenario as they completed the baseline primary outcome measures. The scenario was developed to describe the person's concern about their slow recovery (three weeks later) and their decision to go to see their GP. Participants were randomly allocated to three groups:

  i.  A scan is recommended and the GP provides information about normal scan findings;
 ii.  A scan is recommended (but no information about normal findings is provided);
iii.  The GP provides best practice care (and does not recommend a scan).

Patients who received a scan were further randomised to a type of reporting strategy as detailed in Fig. 1. All participants then completed the primary and secondary outcome measures. Details of the manipulations (and examples) are provided in Supplemental Information 1.

## Measures
### Outcome measures
The primary outcome measure was a composite score derived from three numeric rating scale (NRS) responses (see Supplemental Information 2). The summed outcome score was labelled the Back-Related Perceptions (BRP) score, with higher scores indicating more positive perceptions. We considered that 'patient' reassurance (reduced fear and concern) would be reflected by higher BRP scores. Reassurance is variably defined and measured; we considered that a change in reassurance would be more likely to be detected by a composite outcome measure than a single NRS question.

The two secondary outcome measures were the Tampa Scale of Kinesphobia (TSK-11) (*Tkachuk & Harris, 2012*) and a patient satisfaction rating (see Supplemental Information 3).

### Demographics and baseline screening

Self-report data were collected on age, gender, country of residence, language, and education level attained. Participants completed the Back Pain Attitudes Questionnaire (*Darlow et al., 2014*) (at baseline) and were asked to indicate "*yes*" or "*no*" responses to questions relating to previous back pain, recent back pain, current 'chronic' back pain and history of back scans.

## Data analysis
### Sample size calculation

The study was powered to detect a small effect size, (*Cohen, 1998*) with a power of 80% and an alpha value of 0.05. Assuming a correlation 0.7, the minimum sample size required for each of the three groups was 195 participants.

### Statistical analysis plan

To address each of the primary research aims an analysis of covariance (ANCOVA) was conducted to determine if there was a significant effect of group on the primary outcome (BRP scores) after controlling for the effect of baseline BRP scores. Tukey corrected post-hoc pairwise comparisons (with bootstrapped significance and confidence intervals) investigated whether there was a significant difference between the best practice group and either (or both) of the imaging groups, and between the two imaging groups.

To address the secondary research aims (i) and (ii) we performed one-way analysis of variance (ANOVA) to determine the relationship between the group variable and the secondary outcome measures. If a significant effect was identified, Tukey's post-hoc tests with Bonferroni correction were conducted to identify the between-group differences.

To address secondary research aim (iii) we used ANCOVA to investigate whether there was a significant effect of group after controlling for the effect of baseline BRP scores. The level of significance for this sub-study was set at $P \leq 0.025$ to adjust for multiple comparisons. The level of significance for all other studies was set at $P \leq 0.05$. Statistical analyses was undertaken using SPSS Statistics software (v22.0.0.0, IBM Corporation, New York).

## RESULTS
### Sample characteristics

A total of 788 participants commenced the online study between February and June, 2017. 660 participants completed baseline and post-intervention BRP scores and were included in the analysis. The baseline characteristics of included participants (including mean scores for the Back Pain Attitudes Questionnaire) are presented in Table 1.

### Experimental effects

Table 2 presents mean scores on the primary and secondary outcome measures across experimental groups, and change scores on the primary outcome.

### 'Pre-information' (+ scan) vs 'no pre-information' (+ scan) vs 'best practice'

ANCOVA was conducted to examine the effect of group on BRP whilst controlling for baseline BRP scores. Levene's test and normality checks were carried out and assumptions
**Table 1** Participant characteristics.

| | Total | Group | | |
| | N (%) | No pre-information N (%) | Pre-information N (%) | 'Best practice' N (%) |
|---|---|---|---|---|
| N | 660 | 225 (34) | 216 (33) | 219 (33) |
| Gender | | | | |
| Male | 209 (32) | 68 (30) | 68 (31) | 73 (33) |
| Female | 449 (68) | 157 (70) | 147 (68) | 145 (64) |
| Age category | | | | |
| 18–25 years | 99 (15) | 42 (19) | 29 (13) | 28 (13) |
| 26–35 years | 219 (33) | 68 (30) | 78 (36) | 73 (33) |
| 36–50 years | 210 (32) | 78 (35) | 67 (31) | 65 (30) |
| 51–65 years | 114 (12) | 34 (15) | 37 (18) | 43 (20) |
| 66 years and over | 12 (2) | 3 (1) | 5 (2) | 10 (5) |
| English first language | 561 (85) | 193 (86) | 184 (85) | 184 (84) |
| Education level attained | | | | |
| Did not complete high school | 12 (2) | 5 (2) | 5 (2) | 2 (1) |
| Completed high school | 34 (5) | 8 (4) | 8 (4) | 18 (8) |
| Enrolled in or completed a non- university qualification | 84 (13) | 28 (12) | 34 (16) | 22 (10) |
| Enrolled in or completed an undergraduate degree | 272 (41) | 100 (44) | 82 (38) | 90 (41) |
| Enrolled in or completed a post- graduate degree | 258 (39) | 84 (37) | 87 (40) | 87 (40) |
| Past history of back pain | 622 (94) | 207 (92) | 204 (94) | 211 (96) |
| Recent history of back pain (past 3 months) | 478 (71) | 164 (73) | 160 (74) | 154 (70) |
| Back pain present most days (past 3 months) | 238 (36) | 77 (34) | 84 (39) | 77 (35) |
| Previous back scan | 295 (45) | 96 (43) | 104 (48) | 95 (43) |
| Less than 3 months ago | 31 (5) | 14 (6) | 9 (4) | 8 (4) |
| 3–12 months ago | 49 (7) | 11 (5) | 20 (9) | 18 (7) |
| Between 1 and 5 years ago | 115 (17) | 38 (17) | 40 (19) | 37 (17) |
| More than 5 years ago | 100 (15) | 33 (15) | 35 (16) | 32 (15) |
| | | Group | | |
| Back pain attitudes questionnaire | **Mean (SD)** | **Mean (SD)** | **Mean (SD)** | **Mean (SD)** |
| (Back PAQ) scores | −1.5 (5.9) | −1.5 (5.9) | −1.7 (5.8) | −1.2 (5.9) |

were met. After adjusting for baseline BRP scores there was a significant difference between the three groups in post intervention BRP scores ($F(2, 74) = 10.4, p < 0.001, \eta_p^2 = .04$). Post-hoc pairwise comparisons revealed significant differences between the 'no pre-information' group and the 'best practice' group ($p = 0.001, d = .33$), and between the 'pre-information' group and the 'best practice' group ($p = 0.002, d = .47$). There was no significant difference in BRP between the 2 groups who received scans ($p = 0.202$). For both of the significant comparisons, mean BRP scores (indicating more positive perceptions) were higher for the group who received best practice care.

**Table 2  Primary and secondary outcome scores.**

| Group | N | Baseline BRP Mean (SD) | Post-intervention BRP Mean (SD) | BRP Change score | TSK-11 | Patient satisfaction rating |
|---|---|---|---|---|---|---|
| 1. No pre-information (+scan) | 225 | 18.9 (5.5) | 20.7 (5.8) | 1.8 (4.4) | – | – |
| Standard report ± epidemiological information | 121 | 18.7 (5.6) | 19.8 (6.0) | 1.1 (4.4) | 24.0 (7.0) | 6.2 (2.4) |
| 2. Pre-information (+scan) (Standard report ± epidemiological information) | 216 | 18.5 (5.5) | 20.3 (5.8) | 1.8 (5.0) | 23.0 (7.8) | 6.2 (2.3) |
| 3. 'Best practice' (no scan) | 219 | 19.2 (6.0) | 22.2 (5.4) | 3.0 (4.8) | 22.0 (7.1) | 7.0 (2.4) |
| 4. Standard report | 165 | 18.4 (5.6) | 19.2 (6.0) | 0.9 (4.7) | – | – |
| 5. Standard report + epidemiological information | 174 | 18.9 (5.5) | 21.0 (5.7) | 2.1 (4.7) | – | – |
| 6. Altered summary terminology | 59 | 19.9 (5.6) | 22.2 (5.7) | 2.3 (4.4) | – | – |
| 7. Altered summary terminology + epidemiological information | 45 | 18.4 (4.8) | 21.4 (4.7) | 3.0 (4.8) | – | – |
| 8. Enhanced report (altered summary terminology ± epidemiological information) | 104 | 19.3 (5.3) | 21.8 (6.0) | 2.6 (4.4) | – | – |

One-way ANOVA revealed a significant relationship between group and kinesiophobia ($F(2,553) = 3.0, p = 0.050, \eta_p^2 = .01$). Post-hoc pairwise comparisons revealed a significant difference between the 'no pre-information' group and the 'best practice' group ($p = 0.044$, $d = .29$). Mean kinesiophobia scores were lower for the group who received best-practice care.

There was a significant relationship between group and satisfaction ratings ($F(2,553) = 7.5, p = 0.001, \eta_p^2 = .03$). Post-hoc pairwise comparisons revealed significant differences between the 'no pre-information' group and the 'best practice' group ($p = 0.009, d = .34$), and between the 'pre-information' group and the 'best practice' group ($p = 0.001, d = .33$). Mean satisfaction scores were higher for those who received best practice care.

### Standard reporting vs enhanced reporting

There was a significant effect of group on BRP after controlling for the effect of baseline BRP scores ($F(1,275) = 13.06, p < 0.001, \eta_p^2 = .05$). Mean BRP scores were higher for those who received enhanced reporting.

The individual effects of the two components of enhanced reporting were also examined. When compared with standard reporting, the effect of altering the summary terminology (alone) was significant ($F(1,221) = 7.70, p = 0.006, \eta_p^2 = .03$) as was the effect of both altering the summary terminology *and* including epidemiological information ($F(1,207) = 8.40, p = 0.004, \eta_p^2 = .04$). There was no statistically significant difference between the two versions of enhanced reporting practices ($p = 0.85$).

## DISCUSSION

### Summary

This investigation examined the impact of novel spinal imaging reporting practices on 'virtual' patient BRPs, and provided comparison with best-practice care that does not include imaging. In summary, receiving high quality information (without imaging) resulted in more positive BRPs than receiving imaging results, and was associated with higher satisfaction. Small-to-medium effect sizes indicate the importance of these findings. For patients who were referred for spinal imaging—altering the terminology of the report summary and including epidemiological information on the report improved perceptions, but whether or not information about normal findings was delivered prior to receiving an imaging report had no impact.

### Strengths and limitations

This online study was a novel approach which accessed a large adult population. To our knowledge it is the first investigation to directly compare the reassuring potential of spinal imaging with best practice care, and consider the impact of varied reporting strategies in a randomised design. There are however, some weaknesses with the online scenario-based study design. Participant engagement in the task and understanding of the questions could not be monitored. The usefulness of participants *imagining* that they were the person described in the scenario is uncertain and issues relating to the reliability and validity of this approach are not known. Participants may have had difficulty relating to the patient described, but that 94% reported previous back pain might suggest that they were able to partially identify with the 'patient' experience. Not collecting this information and more detailed data on participant characteristics were shortcomings of this study.

There may also be limitations related to using the primary outcome measure (BRP) as a proxy for reassurance, and the identification of patients' *planned* behaviour (which does not offer information about how someone would *actually* behave in a real situation). In addition, BRP was an outcome measure developed specifically for this study and has not undergone psychometric evaluation. This limitation should be considered when interpreting the strength of this study's findings.

The potential for recruitment bias associated with using the authors' networks to recruit participants was minimised by instructing individuals who were likely to be knowledgeable about 'pain' or 'imaging' to *not* complete the study themselves but to disseminate the advertisement to *uninformed* members of their own networks. However, the high proportion of University educated participants (80%) suggests that the study population is not representative of the general population. The supposed higher health literacy and cognitive ability of study participants is likely to have impacted their interpretation of information and may limit study generalisability.

### Relevance to existing literature

Previous studies have suggested that diagnostic tests for some conditions can offer reassurance, but evidence for the reassuring value of imaging for LBP has been lacking. This study cautiously supports that spinal imaging results—at least when reported in

a 'standard' fashion—are not in themselves reassuring. Findings are also consistent with recent indications that adopting 'enhanced' reporting practices (such as including epidemiological information) can offer benefit (*McCullough et al., 2012*) and supports the view of *Roland & Van Tulder (1998)* that attention to the content and language of spinal imaging reports is warranted. A current randomised trial investigating the inclusion of epidemiological information in lumbar spine imaging reports is likely to further inform the need for practice change (*Jarvik et al., 2015*).

## Implications

The findings of this study have the potential to beneficially inform future health care delivery. General Practitioners are likely to be able to reassure patients with recent-onset LBP more effectively if they offer carefully considered information than if they order a scan. This finding challenges the frequent belief that spinal imaging (for patients with a low likelihood of serious pathology) offers opportunity to alleviate patients' fears and concerns (*Howard & Wessely, 1996*; *Rolfe & Burton, 2013*). Further to this, the suggestion that patients are also likely to be more satisfied with this approach offers that avoidance of a scan is likely to be readily accepted by patients if high-quality information is provided. It is worthwhile acknowledging that this study did not investigate the effect of offering both scans and information. This approach—which is probably common in clinical practice—should be addressed in future studies.

Kinaesiophobia was found to be lower for the 'best practice' group than for the 'no pre-information' group; however, a four-point decrease in TSK-11 scores has been suggested to be required to reflect a clinically important reduction in fear of movement (*Woby et al., 2005*). The two-point between-group difference observed in this study, while statistically significant, may not represent a meaningful difference. Further, although our randomised design and a priori sample size justified not collecting baseline TSK scores, and the BPAQ data showed no difference between groups on that measure at baseline, the between-group differences for TSK and for patient satisfaction were small and we cannot completely eliminate the possibility that they reflect baseline differences. As such, they should be interpreted with caution.

This exploratory study suggests that *simply* offering best practice care is indeed likely to be effective management for recent-onset LBP. That LBP outcomes are not, in general, improving may indicate limited application of the current recommendations or a need to offer clinicians further guidance regarding the provision of quality information. We consider that the information provided to participants in this study (see Supplemental Information 1.) (guided by current literature (*TOP, 2015*; *NICE, 2016*; *Moseley & Butler, 2015*; *Stochkendahl et al., 2017*; *Traeger et al., 2017*)) may not represent routine management and that the potential to improve best practice care currently exists.

This study also suggests that dedicating further attention to the content of spinal imaging reports is likely to be worthwhile. Providing explanations of 'normal' age-related features, carefully adapting summary terminology and accurately interpreting imaging findings offer the potential to positively impact patients' beliefs and concerns, and their subsequent engagement in activity.

### Funding

Emma L. Karran is supported by the Royal Adelaide Hospital Clinical Research Grants (received 2014 & 2015) and the Royal Adelaide Hospital Research foundation Dawes Scholarship. G. Lorimer Moseley is supported by the NHMRC (Principal Research Fellowship ID: 1061279). The funders had no role in study design, data collection and analysis, decision to publish, or preparation of the manuscript.

### Grant Disclosures

The following grant information was disclosed by the authors:
Royal Adelaide Hospital Clinical Research Grants.
Royal Adelaide Hospital Research Foundation Dawes Scholarship.
NHMRC (Principal Research Fellowship): 1061279.

### Competing Interests

G. Lorimer Moseley has received support from: Pfizer Australia, Kaiser Permanente, Results Physiotherapy, Agile Physiotherapy, Workers' Compensation Boards in Australia, Europe and North America, the International Olympic Committee and Port Adelaide Football Club. He receives speaker fees for lectures on pain and rehabilitation. He receives royalties for books on pain and rehabilitation. All other authors declare no conflicts of interest.

### Author Contributions

- Emma L. Karran conceived and designed the experiments, performed the experiments, analyzed the data, wrote the paper, prepared figures and/or tables.
- Yasmin Medalian conceived and designed the experiments, performed the experiments, reviewed drafts of the paper.
- Susan L. Hillier conceived and designed the experiments, reviewed drafts of the paper.
- G. Lorimer Moseley conceived and designed the experiments, reviewed drafts of the paper.

### Human Ethics

The following information was supplied relating to ethical approvals (i.e., approving body and any reference numbers):

This study was approved by the University of South Australia Human Research Ethics Committee (ID 0000035363).

### Data Availability

The raw data has been provided as a Supplemental File.

### Supplemental Information

Supplemental information for this article can be found online at http://dx.doi.org/10.7717/peerj.4151#supplemental-information.

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
