# Peer review of "The impact of choosing words carefully: an online investigation into imaging reporting strategies and best practice care for low back pain"

_PeerJ, doi:10.7717/peerj.4151_

## Round 0.1 · original submission · Major Revisions

Thank you for this interesting article on imaging reporting strategies and best practice care for low back pain. It would be helpful to include information regarding the validity of using simulation patient participants as well as the reliability and validity of the outcome measures utilized.

Reviewer 1 ·

Basic reporting

This article is generally well written, appropriately referenced, conforms to PeerJ standards for structure, and has clear tables.

There are three areas where I believe the basic reporting could be improved:

1) The way in which the intervention groups are described is frequently confusing, both within the abstract and the paper itself, for example “to compare the effect of receiving imaging results with best practice care (without imaging)”. I suggest rewording this with something along the lines of ‘to compare the effect of receiving imaging results with the effect of receiving best practice care that does not include imaging’. It may be helpful to provide a numbered list of the primary aims in the same way the secondary aims are presented.

2) The last sentence of paragraph four in the introduction “Also of interest is evidence that providing patients with information about normal results - prior to their receipt of their own results - may optimise the reassuring potential of normal findings” is discordant with the previous paragraph that suggests imaging is not reassuring. I suggest rewriting the sentence.

3) It would be useful to have exemplars of the pre-information, the normal scan report, the epidemiological information, and the enhanced report that were provided. These could be added to supplement 4 or 7.

Experimental design

I have two key queries in relation to study design that it may be helpful for the authors to address in the discussion:

1) Why was a best practice care plus imaging group was not included?

Although I am not a proponent of scanning for reassurance, I could imagine that if I was I might also consider that I do this in addition to providing best practice care (excluding the explanation that scans are not helpful). This means that I could look at these results and see that best practice care was well received, but also that scores in the scan groups appeared to improve rather than deteriorate after receiving scan reports, so there is a possibility that receiving both of these interventions could be helpful. This dichotomy is carried over to the discussion where it is inferred that GPs choose between either scanning or providing explanation, rather than being able to provide both.

2) Why was the TSK-11 was not completed at baseline?

With no baseline scores it cannot be assumed that the small differences between group found post-intervention are related to the intervention. It is also uncertain whether these between group differences are meaningful (given a four-point change in TSK-11 score is needed to identify an important decrease in fear of movement (Woby et al. 2005) ).

I have some additional queries and suggestions for the authors’ consideration.

3) In terms of participants, were health professionals and other academics included in the authors’ networks and eligible to participate? This may have biased the results as these people may share similar views about imaging to the authors. The risks and benefits of using these networks could be discussed in the strengths and limitations section.

4) Was there any assessment of the psychometric properties of the BRP (which I assume was developed for this study). These could strengthen interpretation based on changes in this measure.

5) When and why was the Back Pain Attitudes Questionnaire completed and what were these data used for? These data are not currently presented.

Validity of the findings

The data generally appear to be robust, however, I am not qualified to comment on the statistical analysis.

1) It would be useful to have mean change scores and 95% CIs presented for the ANCOVA analyses.

2) The sentence “This study provides evidence that simply offering best practice care is indeed effective management for recent-onset LBP” considerably overstates the results from this study as no one with recent onset LBP was included in the study and no clinical outcomes were measured.

Additional comments

Thank you for asking me to review this paper. It is an interesting study design and makes innovative use of technology to explore a hypothesis in a large sample of people. The paper is generally well written and related to current literature and thinking. Given the design, my impression is that the results can only be exploratory and the strength of these is overstated at times in the discussion.

Overall, I think that this paper makes a useful contribution to the knowledge base, and I am supportive of its publication, however, there are a number of places where I think it may be challenged and these may benefit from being addressed.

·

Basic reporting

Overall the article demonstrated a clear, professional and unambiguous English language.In order to improve the clarity I suggest that the author rewords line 5 to 8 in the background section to make the aim less ambiguous. The sentence line i282 is rather awkward and should be restructured,.
Appropriate literature reviews, references and context of this novel work in order to address gaps in the literature, Good links to the high prevalence of degenerative disease in asymptomatic individuals and potential for harmful misinterpretation of imaging. In order to provide more context on the role of imaging I would suggest referring to current guidelines for the use of imaging in LBP to identify cases where imaging is indicated and when it is not. Also add a line to clarify that most cases of LBP are self-limiting and subside within a particular timeframe and does noit require investigations..
Professional article structure, tables and raw data appears to be accurate and clear.
Relevant results were given to confirm the hypothesis

Experimental design

This study is an original primary research within the scope of the journal.
Research aims and secondary aims were well defined, relevant & meaningful. It is stated how research fills an identified knowledge gap. You did not define the research question but identified the knowledge gap being investigated and the study contributes to filling that gap.
Appropriate and novel investigation performed to a high ethical standard. simulated patient, 3-armed, randomised online experiment.

However, how valid and reliable is his simulated patient approach compared to real life patients this needs to be justified. Are there any reviews or studies looking at this? If so I would add to the methodology section. You mentioned that many adult volunteers suffered with LBP but did not indicate the severity or disability associated with this.. You could have included information on participant levels of anxiety or depression, work status and hobbies since these factors could have affected the results. Perhaps add more detail regarding the back pain questionairre and how that impacted upon the results.
Methods described with sufficient detail to replicate the study.

Validity of the findings

Data is robust overall and clearly presented, a sample size calculation was performed the dtests utilised were statistically sound. The participants were randomised, and blinded to the other interventions..

The Primary outcome measure used was the BPS Line 149. Has the validity and reliability of this measure been established? If so it needs to be stated. What was the authors rationale for combining the 3 numerical rating scores. This needs to be stated.
The data on which the conclusions are based are provided and is statistically sound.. Appropriate discussions made where rationale & benefit to the literature and patient management in primary care is clearly stated.

Conclusion are well stated, linked to original research aims & limited to the authors supporting results and the limitations of this study using simulated patients has been addressed.

Additional comments

This is an interesting and novel study with acknowledged limitations but has a sound conclusion and clear implications for primary care practice in the management of low back disorders.

---

## Round 0.2 · accepted · Accept

Thank you for your updated version of your manuscript. It appears you have been responsive to the questions provided.

Reviewer 1 ·

Basic reporting

No comment

Experimental design

No comment

Validity of the findings

No comment

Additional comments

The authors have appropriately addressed the comments and suggestions.